# Standalone Sensors System for Real-Time Monitoring of Cutting Emulsion Properties with Adaptive Integration in Machine Tool Operation

**DOI:** 10.3390/s23135794

**Published:** 2023-06-21

**Authors:** Jozef Peterka, Frantisek Jurina, Marek Vozar, Boris Patoprsty, Tomas Vopat, Vladimir Simna, Pavol Bozek

**Affiliations:** Institute of Production Technologies, Faculty of Materials Science and Technology in Trnava, Slovak University of Technology in Bratislava, 812 43 Bratislava, Slovakia; frantisek.jurina@stuba.sk (F.J.); marek.vozar@stuba.sk (M.V.); boris.patoprsty@stuba.sk (B.P.); tomas.vopat@stuba.sk (T.V.); vladimir.simna@stuba.sk (V.S.); pavol.bozek@stuba.sk (P.B.)

**Keywords:** cutting fluids, probe, concentration measurement

## Abstract

This paper presents a novel cutting fluid monitoring sensor system and a description of an algorithm framework to monitor the state of the cutting emulsion in the machine tool sump. One of the most frequently used coolants in metal machining is cutting emulsion. Contamination and gradual degradation of the fluid is a common occurrence, and unless certain maintenance steps are undertaken, the fluid needs to be completely replaced, which is both un-economical and non-ecological. Increasing the effective service life of the cutting emulsion is therefore desired, which can be achieved by monitoring the parameters of the fluid and taking corrective measures to ensure the correct levels of selected parameters. For this purpose, a multi-sensor monitoring probe was developed and a prototype device was subsequently created by additive manufacturing. The sensor-carrying probe was then placed in the machine tool sump and tested in operation. The probe automatically takes measurements of the selected cutting emulsion properties (temperature, concentration, pH, level height) in set intervals and logs them in the system. During the trial run of the probe, sensor accuracy was tracked and compared to reference measurements, achieving sufficiently low deviations for the purpose of continuous operation.

## 1. Introduction

Sensors can provide real-time information about the machining process, such as temperature, vibration, and tool wear [1], as well as gather data about cutting fluids or cutting emulsions. Although modern machining methods tend towards dry machining [2,3] or machining with a minimum amount of cutting fluids [4], on the other side in conventional machining operations [5,6,7], cutting fluids play a critical role in reducing tool wear, improving the surface finish, and maintaining workpiece accuracy. However, the use of cutting fluids can lead to environmental and health issues, and sensor technology has been proposed as a means of reducing the negative impacts of cutting fluids’ use. To achieve this from a technological point of view, it is important to measure the selected properties of cutting emulsions, preferably in real time. In the field of evaluation of cutting fluids, specifically cutting emulsions, it is possible to monitor many properties of these emulsions. Especially from a technological point of view, the following properties are decisive and most observed: concentration, pH, and temperature.

### 1.1. Concentration Sensors for Cutting Fluids and Cutting Emulsions

There are several types of cutting fluid concentration sensors available for use in machining processes.

One example is the refractometer, which measures the refractive index of a solution and correlates it with the concentration of the cutting fluid [8]. Refractometers are simple and easy to use, with a measurement range of 0 to 100% concentration and a resolution of 0.1% concentration. They are also highly accurate, with an error of less than 1% concentration [9] or up to ±0.1% [10]. They are highly durable and can be used in high-temperature and high-pressure environments [10].

Another type of cutting fluid concentration sensor is the conductivity sensor, which measures the electrical conductivity of a solution and correlates it with the concentration of the cutting fluid [11]. Conductivity sensors are highly sensitive and have a wide measurement range, with a range of 0.1 to 10,000 μS/cm. They are also resistant to fouling and can be used in high-temperature applications [12]. Electrical conductivity sensors are highly accurate, with a measurement range of 0–30% concentration and an accuracy of up to ±0.05%. They are also highly stable and have a long lifespan [13].

In addition to refractometers and conductivity sensors, other types of cutting fluid concentration sensors have been developed for use in machining processes.

These include spectrophotometers, which measure the absorbance of a solution at a specific wavelength and correlate it with the concentration of cutting fluid [14,15], and ultrasonic sensors, which use ultrasonic waves to measure the density of a solution [16], and respectively the speed of sound waves passing through the cutting fluids [17], correlating it with the concentration of the cutting emulsion.

Cutting emulsion concentration sensors are used to measure the concentration of cutting fluids in machining processes. The concentration of cutting fluids can affect their performance in machining, as well as the life of the cutting tool and the quality of the machined surface. Therefore, monitoring the concentration of cutting fluids is essential for optimizing machining conditions and maintaining quality control.

Overall, cutting emulsion concentration sensors are an essential tool for measuring the concentration of cutting fluids in machining processes. While refractometers and electrical conductivity sensors are the most commonly used types of cutting emulsion concentration sensors, new types of sensors, such as ultrasonic sensors and spectrophotometers, offer improved accuracy and reliability for measuring cutting emulsions’ concentration. The use of cutting emulsion concentration sensors in combination with other sensors, such as pH sensors, viscosity sensors, and temperature sensors, can provide a comprehensive picture of the cutting fluids’ performance and help optimize machining operations.

### 1.2. pH Sensors for Cutting Fluids and Cutting Emulsions

Cutting emulsion pH sensors are used to measure the acidity or basicity (pH) of cutting emulsions, which are a type of cutting fluid used in machining processes. The pH of cutting emulsions can affect the performance of the cutting fluid and the quality of the machined surface. Therefore, monitoring the pH of cutting emulsions is essential for optimizing machining conditions and maintaining quality control, and preventing corrosion.

The most common type of pH sensor used for measuring cutting fluid pH is the glass electrode sensor. When the glass electrode comes into contact with a solution, it generates a voltage proportional to the pH of the solution. The reference electrode maintains a constant voltage, allowing for accurate measurement of pH [18].

These types of pH sensors are composed of a glass membrane that is sensitive to changes in pH and a reference electrode that maintains a stable potential. pH electrodes are simple and easy to use, with a measurement range of pH 0–14. They are highly durable and can be used in high-temperature and high-pressure environments [19], and they are highly accurate and reliable, with an error of less than 0.1 pH units [20], and a resolution of 0.01 pH units [21] or an accuracy of up to ±0.05 pH units [19], respectively.

However, glass electrodes can be affected by temperature changes, protein contamination, and other factors that can cause drift in pH readings [22].

Another type of cutting emulsion pH sensor is the colorimetric pH sensor. Colorimetric pH sensors work by changing color in response to changes in the pH, and the color change is measured using a colorimeter or spectrophotometer. Colorimetric pH sensors are highly sensitive, with a measurement range of pH 4–9 and an accuracy of up to ±0.05 pH units. They are also highly stable and have a long lifespan [23].

In addition to pH electrodes and colorimetric pH sensors, other types of cutting emulsion pH sensors have been developed for use in machining processes. These include ion-selective electrodes, which measure the concentration of specific ions, such as hydrogen ions (H^+^) in the cutting emulsion, and correlate it with the pH [24], and fiber optic pH sensors, which use the change in light transmission through a fiber optic cable to measure changes in pH [25].

Overall, cutting emulsion pH sensors are an essential tool for measuring the pH of cutting emulsions in machining processes. While pH electrodes and colorimetric pH sensors are the most commonly used types of cutting emulsion pH sensors, new types of sensors, such as ion-selective electrodes and fiber optic pH sensors, offer improved accuracy and reliability for measuring the cutting emulsion pH. The use of cutting emulsion pH sensors in combination with other sensors, such as concentration sensors, viscosity sensors, and temperature sensors, can provide a comprehensive picture of the cutting fluid performance and help optimize machining operations.

### 1.3. Temperature Sensors for Cutting Fluids and Cutting Emulsions

Cutting emulsion temperature sensors are used to measure the temperature of cutting emulsions, which are a type of cutting fluid used in machining processes. Monitoring the temperature of cutting emulsions is essential for optimizing machining conditions, maintaining quality control, and ensuring the safety of the machining process. The temperature of cutting fluids can affect their performance in machining, as well as the life of the cutting tool and the quality of the machined surface. Therefore, monitoring the temperature of cutting fluids is essential for optimizing machining conditions and maintaining quality control.

One common method for measuring the temperature of cutting emulsions is using thermocouples. Thermocouples are composed of two different metals that produce a voltage in response to changes in temperature. The voltage is then converted to temperature using a calibration curve. Thermocouples are highly accurate, with a measurement range of −200 °C to +1800 °C [26] or −270 °C to 1750 °C [27], and an accuracy of up to ±0.1 °C. They are also highly durable and can be used in high-temperature and high-pressure environments [26].

Another type of cutting emulsion temperature sensor is the resistance temperature detector (RTD). RTDs work by measuring the change in electrical resistance of a metal wire as the temperature changes. The resistance is then converted to temperature using a calibration curve. RTDs are highly accurate, with a measurement range of −200 °C to +850 °C and an accuracy of up to ±0.01 °C [28] or ±0.05 °C [29]. They are also highly stable and have a long lifespan [28].

In addition to thermocouples and RTDs, other types of cutting emulsion temperature sensors have been developed for use in machining processes. These include infrared thermometers, which measure the temperature of a surface by detecting the infrared radiation emitted by the surface [30,31], thermistors, which measure the change in resistance of a ceramic material as the temperature changes and correlate it with the temperature of the cutting fluid [32], and fiber optic temperature sensors, which use the change in light transmission through a fiber optic cable to measure changes in temperature [33].

Overall, cutting emulsion temperature sensors are an essential tool for measuring the temperature of cutting emulsions in machining processes. While thermocouples and RTDs are the most commonly used types of cutting emulsion temperature sensors, new types of sensors, such as infrared thermometers and fiber optic temperature sensors, offer improved accuracy and reliability for measuring the cutting emulsion temperature. The use of cutting emulsion temperature sensors in combination with other sensors, such as concentration sensors, viscosity sensors, and pH sensors, can provide a comprehensive picture of the cutting fluid performance and help optimize machining operations.

Above, an overview of different types of sensors was presented for the properties of the cutting emulsion selected by us—concentration, pH, and temperature. Of course, there are many other types of sensors, the description of which would be very extensive. From this review of our proposal, it follows that the selection criteria, which we used in the next section, are decisive for the selection of sensors.

## 2. Materials and Methods

### 2.1. General Description of the Probe

There were four basic criteria set based on the review of applicable industrial systems currently available as well as the literature review. These criteria were ease of use, compact dimensions, number of tracked parameters, and minimal final unit price. From the standpoint of ease of use, the system needs to be designed in such a way that it can be used even by an operator without extensive knowledge of cutting fluids. Considering the system dimensions, keeping the measurement components as compact as possible is desirable so that the system can be applied on as many machines as possible, since the machine sump space can vary. Factors of the number of tracked physical and chemical parameters are closely linked to the unit price; therefore, it was important to select the most important indicators of the cutting fluid state while keeping the price reasonably low.

The conceptual design of the system is composed of hardware and software parts. The hardware part, which contains the sensor probe and the display unit, is responsible for data collection and their basic overview. The software part consists of a web application which is used to analyze the collected data and view scheduled maintenance tasks.

The probe itself is meant for data collection of selected physical and chemical characteristics of the cutting fluid. All the sensors are placed within the probe, and its compact dimensions allow for it to be placed in the machine sump, where it floats on the surface of the cutting fluid. Measured data are transferred utilizing a wireless network. An illustration of the probe placement in the CNC machining center sump can be seen in Figure 1.

The probe body consists of the housing and the top cover. When specifying the dimensions and designing the shape, the number of sensors and their correct placement was the main focus of the process. Moreover, it was necessary to ensure sufficient buoyancy of the probe body, as it was meant to be floating on the surface of the cutting fluid during its operation. The probe body in our case was fabricated by FDP (fused deposition modeling) 3D (three-dimensional) printing. ABS material was used to produce the probe, which is the standard material supplied for the 3D printing method. The dimensions of the probe were as follows: diameter D = 140 mm and height = 90 mm [34]. An advantage of this method of manufacturing was that the Qi charging coil could be incorporated into the lower part of the probe, allowing for wireless charging by placing the probe on the charging pad. A rechargeable lithium-polymer battery with a capacity of 750 mAh was used to power the probe. Sensors and other probe components had to be placed in the body in such a way as to not shift the center of gravity, which would result in non-uniform buoyancy. However, each sensor had to be placed in a way that would not impede its functionality with regard to the measurement of any given cutting fluid property. For this reason, the combined sensor for the measurement of fluid concentration and temperature, which is the heaviest component of the probe, was placed in the lowest area of the probe body, facing downward. This ensures that the debris and sediments do not accumulate on the sensor, keeping it accurate and maintenance-free. The sensor for the pH measurement was positioned on the side of the probe. The fluid level sensor was located on the top of the device. A basic overview of the probe features is illustrated in Figure 2.

The custom-made electronic circuit board placed in the middle part of the probe is the core component of the probe with all the sensors connected to it, acting as a bus to collect all the measured data. As the circuit board itself is not waterproof, the interior of the probe had to be made airtight, using a high-grade silicone sealant for all points of contact with the cutting fluid.

### 2.2. Software and Communication

The main component of the electronic circuit board is the SoC ESP32 (Espressif systems, Shanghai, China). This chip combines a 2-core ARM processor, an I/O converter for communication with peripheral devices—sensors, wireless and Bluetooth connectivity, and program memory. The microcontroller is responsible for the collection and transmission of the measured data to the server, where they are evaluated further. On top of the aforementioned component, there are other subsections present on the circuit board, such as the power management section, voltage references for each measurement sensor, an antenna for wireless connectivity, and signal LED indicators.

The probe’s firmware running on the ESP32 microcontroller (Espressif systems, Shanghai, China) was created in the Wiring programming language for Arduino IDE 2.1.0. Its structure consists of loading saved constants, measurement initialization, relay of measured data to the server, and switching to the power-saving mode until the next measurement cycle.

A web application for the management of measured data and their evaluation was designed with a simple and accessible user interface. The purpose of the application was to have a simple interface for cutting fluid management that can be scaled as needed. There are multiple levels of the application structure, with multiple companies, different workplaces, and a number of machines being tracked. For each machine, there are up-to-date measured values plotted into graphs. Upper and lower limits for each parameter can be set, depending on the type of cutting fluid used and the desired operation. When these limits are exceeded, an alert is sent to the operator, requiring a maintenance action to be carried out. Probe calibration and cleaning can also be scheduled through the application, as well as charging. Moreover, measured parameters and the frequency of measurement can be changed depending on the requirements.

### 2.3. List of Probe’s Sensors and Their Technical Specifications

Based on the literature review and previous experiments [35], a set of specific sensors was chosen to be placed in the probe and carry out measurements:Digital refractometer for measurement of concentration.Shock-proof ISFET (ion-sensitive field-effect transistor) for pH measurement.NTC (negative temperature coefficient) thermistor for temperature measurement.Optical distance sensor for measurement of the fluid level.

The sensor for the measurement of concentration is composed of mechanical parts—sensor housing and semi-permeable optical prism, source of light, and detector of the amount of reflected light—with designation Hamamatsu S111-06-10 (Hamamatsu Photonic, Shizouka, Japan). The measurement of concentration was conducted as follows. At first, the light source (LED) emitting light onto the semi-permeable mirror is initialized. Based on the index of refraction of the measured fluid, a certain amount of light is reflected back to the detector. The number of pixels receiving light is transmitted to SoC ESP32 (Espressif systems, Shanghai, China), and based on a mathematical renumeration, it is possible to obtain the percentual level of the fluid concentration.

The sensor for pH measurement is composed of an ISFET measurement component with designation Senstron A120-001 (Sentron Europe, Leek, The Netherlands), with optimized dimensions so that it can fit inside the probe housing. Measurement begins with initialization of the entry module of the ISFET pH measurement component. Subsequently, a frontend module conducts the pH measurement, after which the obtained value is sent to SoC ESP32 (Sentron Europe, Leek, The Netherlands). Alongside the pH measurement, the temperature is measured as well, using the PT1000 thermistor (Shenzhen Huayaya Technology Co., Ltd., Shenzhen, China).

The optical distance measurement sensor with designation RFD77402 (Sparkfun Electronics, Niwot, CO, USA) utilizes a laser beam to measure the time of flight, based on which the distance is calculated.

To verify the measurement results of the probe, reference measurements were conducted manually by hand-held devices in the same intervals as the automated probe measurements. The specifications of these devices were as follows:Hand-held optical refractometer RHB18ATC (Yhequipment Co., Shenzhen, China).Combined measuring instrument, Testo 206-pH1 (Testo, West Chester, PA, USA), for pH and temperature measurements.Machine sump fluid volume indicator.

### 2.4. Trial Operation

After the prototype probe was manufactured and assembled, it was necessary to verify the functionality and accuracy of the sensors and other components of the probe. A test run of the probe was accomplished via the following steps:Verification of the sensors’ functionality.Verification of the communication between the device and the server.Verification of the data input into the database.Verification of the web application’s functionality.Sensor calibration.Operation in real conditions.

The functionality of each sensor in the probe was tested by connecting the probe to the computer through the RS232 interface to USB. Testing firmware was loaded in this mode, the purpose of which was to verify if all the sensors are active and working, carry out measurements, and send back the measured data through the serial port into the computer, where they were saved in a plain text file. All the measured data were raw, meaning that analogue sensors’ output were voltage values and digital sensors’ output were logic level values. After the test was successfully performed, working firmware was loaded to verify the communication between the device and the web application.

In order to test the communication between the probe and the server, it was necessary to establish a wireless connection between the device and the Wi-Fi access point. After establishing a communication, the probe transmits measured values in the format of a GET request, which is processed by the server and saved into a database. After confirming a successful data entry, a new setup sequence is sent back to the probe from the server. The setup sequence contains data such as the time of the next measurement and variables important for the correction of measured values. Once the measured values are saved on a server, they can be processed. For processing the data, a.php script is used, that writes the values into a database and automatically sends back the setup sequence.

After obtaining the measured data from the device, it was possible to verify the functionality of the web application. The verification procedure consisted of checking the loading of data from the database and the subsequent display in selected parts of the application. Furthermore, the functionality of the event log and the generation of measurement logs were checked.

### 2.5. Calibration of Probe Sensors

The next step after the verification of probe functionality was calibration of each sensor. For proper calibration, it was necessary to take into consideration the type of cutting fluid used as well as the minimum and maximum levels of cutting fluid in the machine sump. Calibration of the digital refractometer required preparation of multiple calibration solutions with different concentrations of cutting fluid. Concentration values of these solutions were in the range of 1% to 15%, with a 1% value difference between each solution sample. For the preparation of the samples, a base concentrate of Castrol cutting fluid with designation Alusol ABF 10 (Castrol Classic Oils, Cambridge, UK) was mixed with deionized water. The functioning mechanism of the digital refractometer is similar to the hand-held refractometer, with a couple of differences.

The main difference was the source of light used, which was high-luminosity LED, and as an observational element, an optoelectrical photosensor with 120 light-sensitive pixels was utilized. The working principle of this sensor is illustrated in Figure 3.

The process of calibration was as follows: A cutting fluid sample with a specific concentration was applied on the refractometer prism, and the voltage value of each pixel was recorded. This process was repeated for every concentration sample solution. The values of the concentrations of the samples were paired with the recorded voltage values of the lit refractometer pixels. Figure 4 is an example of a digital oscilloscope reading for 1% and 100% concentration solutions. It can be observed that the higher the concentration of the solution, the more unlit pixels displaying low-voltage values there were. Based on this reading, the value of the measured concentration can be calculated.

The calibration of the fluid level in the machine sump required setting the correction coefficient to accurately measure the fluid level height. The sensor output is a value of distance in millimeters, measured from the face of the sensor to the nearest surface reflecting back. The distance sensor is based on the optical principle, calculating the distance by measuring the time it takes an emitted optical beam to be received back. When the probe is placed in the machine sump, the optical beam is reflected from the top cover of the tank. When the value of distance is measured, a percentual value, representing how full the sump is, is calculated based on the parameters of the minimum and maximum levels of cutting fluid in the machine sump.

The sensor for the pH measurement was calibrated in a similar way as the concentration sensor. Calibration solutions, called pH puffers, were prepared, with pH values of 4, 7, and 10. The reference electrode and the ISFET pH sensor were submerged into the first puffer, and the measured value of voltage was recorded. The same steps were followed with the rest of the pH puffers. Measured voltage values were assigned corresponding pH values, which allowed for measuring the complete range of the pH parameter.

The temperature sensor in the probe consists of a voltage divider containing a series-connected NTC thermistor and a resistor with a value of 100 KΩ.

Calibration of the temperature sensor consisted of entering the constants for the used NTC thermistor. The constants that need to be entered for the correct temperature calculation are:Reference resistance of the thermistor.Temperature for reference resistance.Beta factor.Reference supply voltage.

After entering the correct values, the accuracy of the measurement may be affected by an unstable reference voltage or improper placement of the thermistor. The thermistor touches the metal part of the concentration sensor, which is in direct contact with the cutting fluid. Sufficient heat transfer between the NTC thermistor and the metal part of the concentration sensor is ensured by the application of a thermal paste with high thermal conductivity.

## 3. Results

Trial operation of the probe was accomplished at the Center of Excellence 5-axis Machining Laboratory of the Faculty of Materials Science and Technology, Slovak University of Technology, Bratislava. For a testing run of the probe, a 5-axis milling center DMG DMU 85 monoBlock was selected due to the type of cutting emulsion used, as well as the frequent operation of this machine tool, which ensures sufficient circulation of the cutting fluid. Blaser Blasocut BC 25 MD water-miscible cutting emulsion concentrate was used for this machine.

Limit thresholds for tracked parameters were set in the probe’s software and are listed in Table 1.

All the tracked parameters should be kept within these limits. In case any of the measured values exceeded these limits, an alert specifying a maintenance action would be displayed in the application. The measured data listed in the following plots were gathered over a period of two weeks. No reference measurements were taken during the weekends, hence the lack of data to complement the automated measurements that took place during the weekends. These missing measurements are displayed as dashed lines interpolated between the measured values. The machine tool runtime was approximately 12 h per day, with the exception of two days of downtime during the weekends. Different machining tasks, ranging from conventional machining of aluminum alloys to high-feed machining of nickel-based super-alloys, were conducted during the machine operation, accounting for some of the parameter volatility recorded by the probe measurements. Previous replacement of cutting fluid took place approximately three months before the start of the track, with periodic maintenance actions such as the addition of deionized water or oil concentrate.

In Figure 5, values of concentration measured by the probe are plotted. Deviation of the sensor, as specified by the manufacturer, was ±0.2%. Deviations from the reference measurements are displayed as a separate data line. It is apparent from the graph that while the measured values were within the deviation range specified by the manufacturer of the sensor, the deviation from the reference values was present for almost all measured values. This is mainly due to the high precision achieved for the reference measurement. The measured values of the concentration had increasing character due to the decreasing fluid volume caused by water evaporation.

Values of measurement for the percentage of the sump fill level are plotted in Figure 6. The deviation of the optical distance sensor was ±1 mm. The error of the reference measurement was the same, and therefore, all the measured values were within an achievable precision tolerance. The level of the cutting fluid in the machine sump was decreasing because of the aforementioned water evaporation. On the last day of the trial run, a maintenance action was performed based on an issued warning about the cutting fluid volume decreasing below the set threshold value. Deionized water mixed with cutting fluid concentrate was added to the machine sump, as shown by a subsequent measurement, number 29.

Temperature measurement results are listed in Figure 7. Deviations of both the probe sensor and the reference method were ±1 °C, so all the measured values except one outlier were within the deviation range. While temperature plays a significant role in the condition of the cutting fluids, during machining operation, it can accumulate heat from the separated chip, which can cause spikes in the cutting fluid temperature. Ambient air temperature also plays a role, and air temperature regulation in the laboratory where the machine tool is located was rather limited. Therefore, it is very difficult to regulate the cutting fluid temperature without a dedicated cutting fluid heat exchanger. A maintenance alert was issued on the 10th day; however, after the reference measurement, it was concluded that the set threshold for the maximum temperature was not reached. It is possible that the temperature within the cutting fluid was not homogenous at the time.

The sensor for the measurement of pH in the probe had the best accuracy out of all used sensors, with its deviation being ±0.01 pH point. However, there were still differences observed from the reference measurement values, even though the deviation values were considered low enough for the purpose of sufficient real-time monitoring of the cutting fluid. Recorded pH values are shown in Figure 8. Spikes in the pH values were most likely caused by a combination of the declining cutting fluid volume because of water evaporation and the admixture of tramp oils from the machined parts. During the weekends when the machine tool was not in operation, the pH values seemed to stabilize as the cutting fluid became briefly stagnant in the machine sump.

In case the monitored fluid parameters diverge from the set range, the wireless system automatically sends a notification to the machine operator, who should then carry out corrective maintenance steps. This can lead to a significant extension of the effective service life of the monitored cutting emulsion.

On the 15th day, a maintenance alert was issued because the probe measured values exceeding set limits, and a maintenance action was carried out. Cutting oil concentrate mixed with deionized water was added to the machine sump. Due to the time required to fill up the machine sump, a measurement run was postponed, to take place afterwards.

## 4. Discussion

The measurement of four key parameters of the cutting fluid by a prototype probe containing multiple sensors was conducted over a period of 15 days. By periodic monitoring of the cutting fluid state, the aim was to maintain and extend the effective life of the cutting fluid, allowing for more economical and ecological production, since the cutting fluid does not need to be repeatedly discarded and refilled.

The trial run of the prototype probe achieved all the objectives that were initially set:Verification of the probe’s full functionality.Verification of the probe’s sensors’ accuracy falling within the acceptable tolerance.Comparison of data measured by the probe to the reference measurements.Verification of the developed web application’s functionality for data-logging.Automated alerts’ functionality verification.

Throughout the operation of the probe for the duration of 15 days, there were no errors or major issues with any part of the probe’s systems. Reference measurements were carried out primarily to obtain accurate data about the cutting fluid parameters using standard devices that are intended for cutting fluid measurements. While the probe was continuously operating, all the sensors were periodically checked for the presence of debris and oil residue. It was found that a tramp oil layer was accumulating on the concentration sensor located on the bottom of the probe. Manual cleaning with alcohol solution temporarily alleviated this issue, as no large deviations or spikes were recorded.

Average deviations of values obtained by the probe’s sensors from the reference measurements were:0.614% for the concentration measurement.0.609% for the sump fill level.0.690 °C for the temperature measurement.0.040 for the pH measurement.

These deviations can be considered acceptable from the standpoint of the tracked parameters and their influence on the cutting fluid condition; however, it is important to note that these measurements were taken over a relatively short period, and it is entirely possible that during a continuous operation for months or years, these values would probably be considerably higher. Further testing of the probe’s long-term reliability is needed to expand and verify the obtained results.

A comparison with commercially available systems for the monitoring and refilling of the cutting fluid is not entirely possible due to the lack of availability of detailed technical specifications of the sensors used for such systems. Moreover, commercially available systems require complex integration into the machine tool systems, which requires partial disassembly of the machine tool and machine sump for the purpose of installing monitoring and/or refilling systems. On the contrary, the proposed probe design can be implemented without any technical intervention and functions independently of the machine tool hardware and software. Commercial systems also require either in-house qualified operators or external technicians for this purpose.

## 5. Conclusions

A novel method for real-time cutting fluid parameters’ monitoring was presented in the paper and the preliminary results of the probe’s testing run were illustrated.

This method (monitoring of cutting emulsions in real time) can be included among the so-called Ecologically Friendly Machining (EcoFriM) methods, which include: dry machining [36,37], machining with a minimum amount of cutting fluids and cryogenic machining [38,39], the use of ecological (biodegradable) cutting media [40], and replacing oil with ecologically acceptable cutting emulsions [5], monitoring the properties of cutting fluids to increase their service life while maintaining the sufficient quality of the machined surfaces.

The proposed sensor system automatically measures, collects, and logs data about the state of the cutting fluid, allowing for continuous monitoring and timely intervention in case the measured parameters do not meet the set criteria. Evaluation of the measurement results and their comparison to reference measurements showed promising reliability and sufficient accuracy of the probe prototype.

The main advantages of the proposed system are the following:Real-time monitoring of cutting fluid parameters.Simple user interface, facilitating ease of use.Automated maintenance action notifications and alerts.Probe can be used with virtually any machine tool with direct access to the machine sump.Configurable properties of the measurement, such as frequency, time, and parameter limits.

While the trial run of the probe’s capabilities achieved success, there are still areas where more development is needed, primarily the connectivity range and further web application extension. Since the probe is enclosed in a machine sump, which is essentially a metal container, the signal strength for the wireless connection might prove to be insufficient depending on the location of the closest wireless access point. Signal amplifier placement near the machine sump could help alleviate this issue. The concentration sensor accumulating the tramp oil layer could be fixed by changing the placement of the sensor to the side of the probe; however, this would require a complete redesign of the probe housing. Another potential issue that may arise is that the placement of the probe in the tank needs to be fixed, as due to the small dimensions of the probe, it can get stuck in constricted parts of the machine sump, which can lead to inaccurate measurements or damage of the probe. Currently, probe functionality is focused on water-miscible semi-synthetic cutting fluids; however, in the future, a probe with improved functionality for the measurement of fully synthetic cutting fluids and cutting oils should be developed.

## 6. Patents

This article stems from the research and development work of the CE5AM (Centre of Excellence of 5-Axis Machining), also in connection with the utility model and patent.

Device for automatic collection of data on selected properties of cutting fluid: utility model application No. 35-2020, date of filing: 24 March 2020, date of publication: 24 March 2021. Bulletin of the Industrial Property Office of the Slovak Republic No. 06/2021, status: valid, registered UI No. 9240, date of notification of UI registration: 28 July 2021. Bulletin of the Industrial Property Office of the Slovak Republic No. 14/2021. Banská Bystrica: Industrial Property Office of the Slovak Republic, 2021. 5p. Available from: https://wbr.indprop.gov.sk/WebRegistre/UzitkovyVzor/Detail/35-2020 (accessed on 21 January 2023).

Device for automatic collection of data on selected properties of cutting fluid: patent application No. 22-2020, date of filing: 24 March 2020, status: published patent application, date of publication of the application: 13 October 2021, Bulletin of the Industrial Property Office of the Slovak Republic No. 19/2021. Banská Bystrica: Industrial Property Office of the Slovak Republic, 2021. 5p. Available online: https://wbr.indprop.gov.sk/WebRegistre/Patent/Detail/22-2020 (accessed on 21 January 2023).

## Figures and Tables

**Figure 1 sensors-23-05794-f001:**
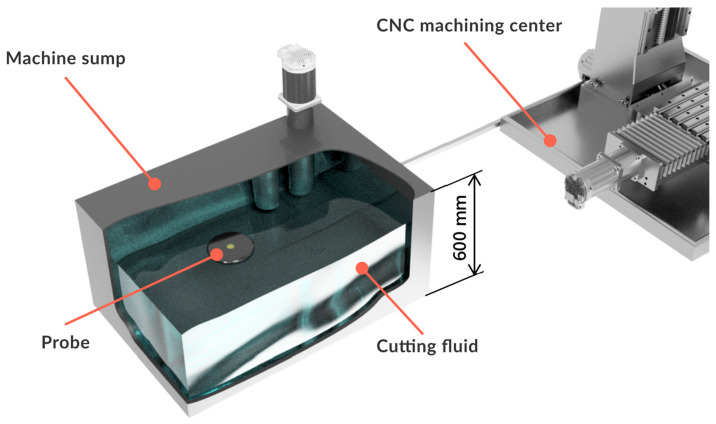
Basic scheme of probe placement.

**Figure 2 sensors-23-05794-f002:**
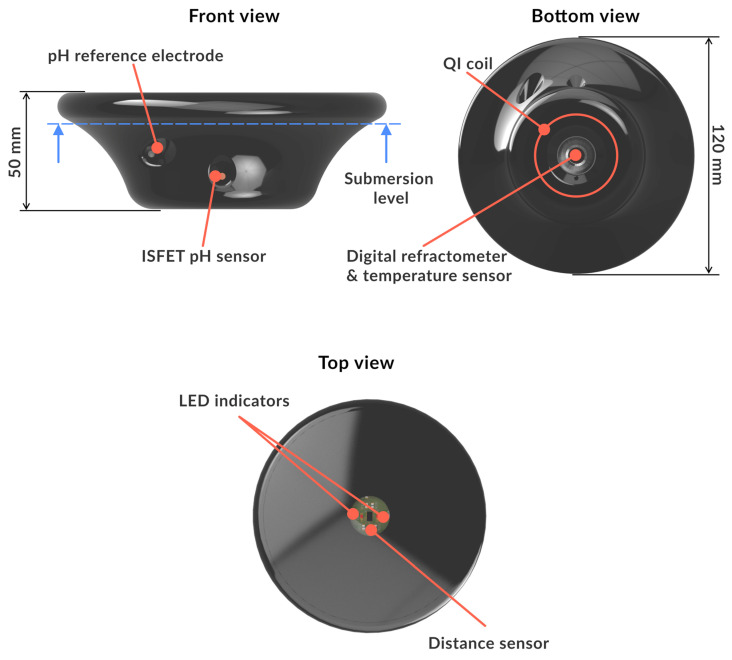
Location of probe’s sensors.

**Figure 3 sensors-23-05794-f003:**
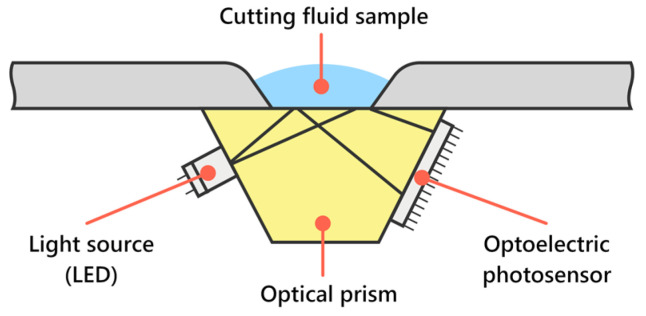
Scheme of the concentration measurement sensor.

**Figure 4 sensors-23-05794-f004:**
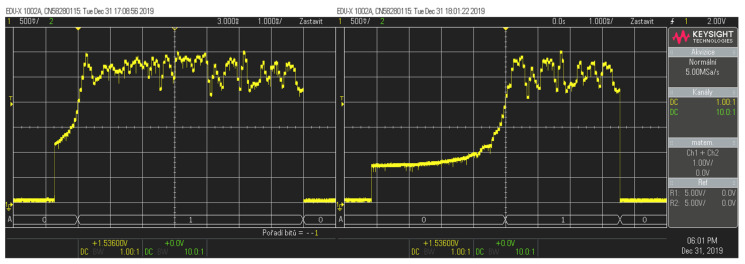
Oscilloscope readings for 1% and 100% cutting fluid concentrations.

**Figure 5 sensors-23-05794-f005:**
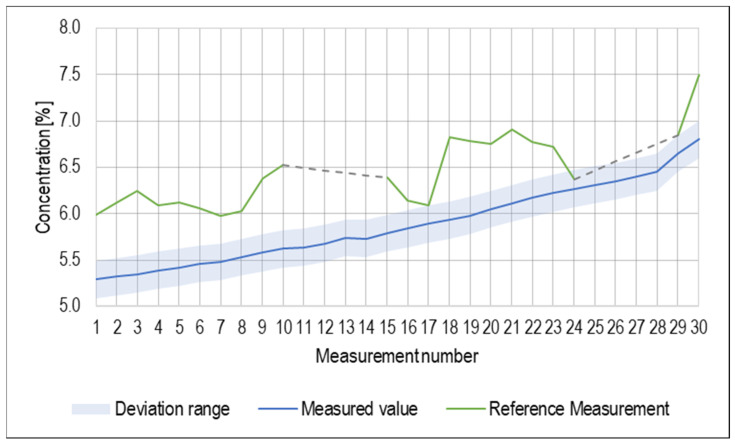
Values of concentration measurement.

**Figure 6 sensors-23-05794-f006:**
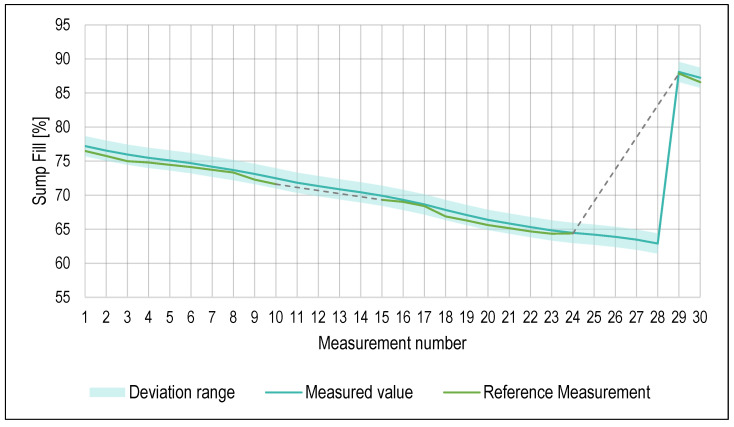
Values of sump fill measurement.

**Figure 7 sensors-23-05794-f007:**
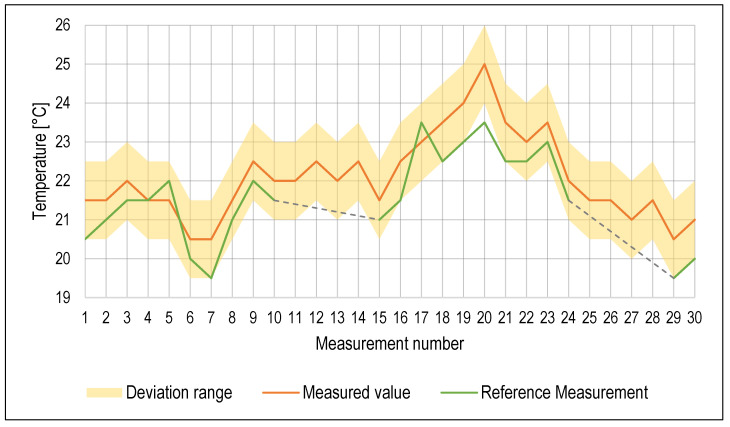
Values of temperature measurement.

**Figure 8 sensors-23-05794-f008:**
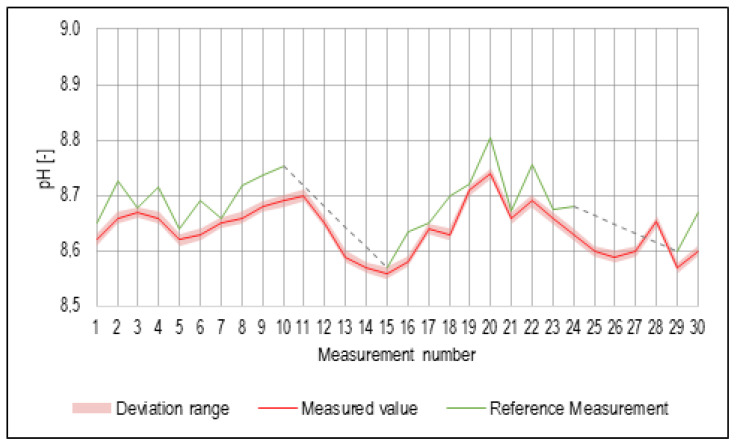
Values of pH measurement. Note: A dashed line means that reference measurements were not performed.

**Table 1 sensors-23-05794-t001:** Thresholds set for a maintenance action alert.

Threshold	Concentration (%)	Sump Fill Level (%)	Temperature (°C)	pH (-)
min	5	65	18	8.5
max	8	95	25	9.4

## Data Availability

Not applicable.

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
