# Peer review of "Standalone Sensors System for Real-Time Monitoring of Cutting Emulsion Properties with Adaptive Integration in Machine Tool Operation"

_sensors, 2023, doi:10.3390/s23135794_

Round 1
Reviewer 1 Report
1. The main findings of this work are not clearly written in the abstract.
2. Figure 2’s caption should come after the figure (line#206).
3. The probe is floating on the coolant. Which part of the probe will be submerged in the coolant? Please label the submerged region in Figure 2 at the side view.
4. Will the electronic circuit board (line#208) be contacted with the coolant? The circuit board is sealed or waterproof? Please briefly explain in line#208.
5. The full definition is not provided in the first use of ISFET (line#241) and NTC (line#242).
6. Please provide the University name in line#364 after the faculty name.
7. Briefly discuss why the measured value continuing drop after 24 when compared to the reference measurement in Figure 6 and the sudden increase at 28.
8. Please correct the number format at the Y–axis in Figure 7. The comma for the number may lead to a misunderstanding of the temperature value.
9. What is the meaning of the dashed line in Figure 8? Please mention this in the manuscript.
10. What causes the higher pH at 20, as shown in Figure 8? Please discuss.
11. The conclusion is not provided in the manuscript.
Minor editing of English language required
Reviewer 2 Report
This paper introduces a sensor system for monitoring cutting fluids, accompanied by an algorithm framework designed to assess the condition of the cutting emulsion within the machine tool sump. Cutting emulsion is widely used as a coolant in metal machining processes. However, it is prone to contamination and gradual degradation over time. If proper maintenance measures are not implemented, complete replacement of the fluid becomes necessary, resulting in significant cost and environmental implications. Therefore, there is a growing need to extend the useful lifespan of cutting emulsion by continuously monitoring its parameters and implementing appropriate corrective actions to maintain optimal levels of selected variables.
I am not sure you followed the format of the Journal, check it
Regarding state of the art: no any reference to the works of Dr. O.Pererira, about MQl or coolants, see https://doi.org/10.1016/j.jclepro.2017.07.078 and many other, some even showing that coolant are a place where organism live. Many works missed indeed. Perhaps that is the reason to select sensors, the work need to complete sensors with the real problems, or it is not acceptable. Patents are OK, but please reduce the section or include the patents about BeCold as well. The style is not logic for a paper; papers must be in the reference list.
Discussion is not in a good style.
In general, paper need a change of style, the introduction is weak, ideas are quire good and results must be shown in paper style.
Figure 1: show some length or scale.
Reviewer 3 Report
1. The authors need to improve the structure of Section 1. It is proposed to issue the titles of subsections with a number. For example "1.1 Concentration sensors for cutting fluids and cutting emulsions", etc. It is also necessary to correct paragraphs spacing in accordance with the rules of the Sensors magazine.
2. Figures 1 and 2 is too dark. Since the authors presented them precisely as schemes, they should be more illustrative. In its present form, it will be difficult for the reader to see the details. It is proposed to improve the quality and clarity of Figures 1 and 2..
3. Figure 2 title should be placed below the figure itself. At the moment, after the picture, there is a part of the text of the article, and only then the name of the Figure, which is most likely a mistake.
4. Need to improve the quality of Figure 4
5. The discussion of the obtained results is not presented in the form in which it is required in journals of this level. A detailed comparative analysis of the obtained results with the results obtained earlier by other authors should be carried out.. The authors must give a detailed comparison, and only then will the reader understand the contribution of the authors to science and scientific novelty, and the new knowledge that has been obtained or the existing ideas that have been developed.
6. In general, the article is interesting and scientifically developed. The article is quite promising and after correction in accordance with the comments may be published.

Round 2
Reviewer 2 Report
Ok